# Reinforcement learning for instance segmentation with high-level priors

## Abstract

Instance segmentation is a fundamental computer vision problem which remains challenging despite impressive recent advances due to deep learning-based methods. Given sufficient training data, fully supervised methods can yield excellent performance, but annotation of groundtruth remains a major bottleneck, especially for biomedical applications where it has to be performed by domain experts. The amount of labels required can be drastically reduced by using rules derived from prior knowledge to guide the segmentation. However, these rules are in general not differentiable and thus cannot be used with existing methods. Here, we revoke this requirement by using stateless actor critic reinforcement learning, which enables non-differentiable rewards. We formulate the instance segmentation problem as graph partitioning and the actor critic predicts the edge weights driven by the rewards, which are based on the conformity of segmented instances to high-level priors on object shape, position or size. The experiments on toy and real data demonstrate that a good set of priors is sufficient to reach excellent performance without any direct object-level supervision.

## 1 Introduction

Instance segmentation is the task of segmenting all objects in an image and assigning each of them a different id. It is the necessary first step to analyze individual objects in a scene and is thus of paramount importance in many computer vision applications. Over the recent years, fully supervised instance segmentation methods have made tremendous progress both in natural image applications and in scientific imaging, achieving excellent segmentations for very difficult tasks (Chen, Wang, and Qiao 2021; Lee et al. 2017).

A large corpus of training images is hard to avoid when the segmentation method needs to take into account the full variability of the natural world. However, in many practical segmentation tasks the appearance of the objects can be expected to conform to certain rules that are known *a priori*. Examples include surveillance, industrial quality control and especially medical and biological imaging applications where full exploitation of such prior knowledge is particularly important as the training data is sparse and difficult to acquire: pixelwise annotation of the necessary instance-level groundtruth for a microscopy experiment can take weeks or even months of expert time. The use of shape priors has a strong history in this domain (Delgado-Gonzalo et al. 2014; Osher and Paragios 2007), but the most powerful learned shape models still require groundtruth (Oktay et al. 2018) and generic shapes are hard to combine with the CNN losses and other, non-shape, priors. For many high-level priors it has already been demonstrated that integration of the prior directly into the CNN loss can lead to superior segmentations while significantly reducing the necessary amounts of training data (Kervadec et al. 2019). However, the requirement of formulating the prior as a differentiable function poses a severe limitation on the kinds of high-level knowledge that can be exploited with such an approach. Our contribution addresses this limitation and establishes a framework in which a rich set of non-differentiable rules and expectations can be used to steer the network training.

To circumvent the requirement of a differentiable loss function, we turn to the reinforcement learning paradigm, where the rewards can be computed from a non-differentiable cost function. We base our framework on a stateless actor-critic setup (Pfau and Vinyals 2016), providing one of the first practical applications of this important theoretical construct. In more detail, we solve the instance segmentation problem as agglomeration of image superpixels, with the agent predicting the weights of the edges in

the superpixel region adjacency graph. Based on the predicted weights, the segmentation is obtained through (non-differentiable) graph partitioning. The segmented objects are evaluated by the critic, which learns to approximate the rewards based on object- and image-level reasoning (see Fig. 1).

The main contributions of this work can be summarized as follows: (i) we formulate instance segmentation as a RL problem based on a stateless actor-critic setup, encapsulating the non-differentiable step of instance extraction into the environment and thus achieving end-to-end learning; (ii) we do *not* use annotated images for supervision and instead exploit prior knowledge on instance appearance and morphology by tying the rewards to the conformity of the predicted objects to pre-defined rules and learning to approximate the (non-differentiable) reward function with the critic; (iii) we introduce a strategy for spatial decomposition of rewards based on fixed-sized subgraphs to enable localized supervision from combinations of object- and image-level rules. (iv) we demonstrate the feasibility of our approach on synthetic and real images and show an application to two important segmentation tasks in biology. In all experiments, our framework delivers excellent segmentations with no supervision other than high-level rules.

## 2 RELATED WORK

Reinforcement learning has so far not found significant adoption in the segmentation domain. The closest to our work are two methods in which RL has been introduced to learn a sequence of segmentation decision steps as a Markov Decision Process. In the actor critic framework of Araslanov, Rothkopf, and Roth 2019, the actor recurrently predicts one instance mask at a time based on the gradient provided by the critic. The training needs fully segmented images as supervision and the overall system, including an LSTM sub-network between the encoder and the decoder, is fairly complex. In Jain et al. 2011, the individual decision steps correspond to merges of clusters while their sequence defines a hierarchical agglomeration process on a superpixel graph. The reward function is based on Rand index and thus not differentiable, but the overall framework requires full (super)pixelwise supervision for training.

Reward decomposition was introduced for multi agent RL by Sunehag et al. 2017 where a global reward is decomposed into a per agent reward. Bagnell and Ng 2006 proves that a stateless RL setup with decomposed rewards requires far less training samples than a RL setup with a global reward. In Xu et al. 2019 reward decomposition is applied both temporally and spatially for zero-shot inference on unseen environments by training on locally selected samples to learn the underlying physics of the environment.

The restriction to differentiable losses is present in all application domains of deep learning. Common ways to address it are based on a soft relaxation of the loss that can be differentiated. The relaxation can be designed specifically for the loss, for example, Area-under-Curve (Eban et al. 2017) for classification or Jaccard Index (Berman, Triki, and Blaschko 2018) for semantic segmentation. These approaches are not directly applicable to our use case as we aim to use a variety of object- and image-level priors, which should be combined without handcrafting an approximate loss for each case. More generally, but still for a concrete task loss, Direct Loss Minimization has been proposed in Y. Song et al. 2016. For semi-supervised learning of a classification or ranking task, Discriminative Adversarial Networks have been proposed as a means to learn an approximation to the loss (Santos, Wadhawan, and Zhou 2017). Most generally, Grabocka, Scholz, and Schmidt-Thieme 2019 propose to train a surrogate neural network which will serve as a smooth approximation of the true loss. In our setup, the critic can informally be viewed as a surrogate network as it learns to approximate the priors through the rewards by Q-learning.

Incorporation of rules and priors is particularly important in biomedical imaging applications, where such knowledge can be exploited to augment or even substitute scarce groundtruth annotations. For example, the shape prior is explicitly encoded in popular nuclear (Schmidt et al. 2018) and cellular (Stringer et al. 2021) segmentation algorithms based on spatial embedding learning. Learned non-linear representations of the shape are used in Oktay et al. 2018, while in Hu et al. 2019 the loss for object boundary prediction is made topology-aware. Domain-specific priors can also be exploited in post-processing by graph partitioning (Pape et al. 2019). Interestingly, the energy minimization procedure underlying the graph partitioning can also be incorporated into the learning step (Abbas and Swoboda 2021; Maitin-Shepard et al. 2016; J. Song et al. 2019).

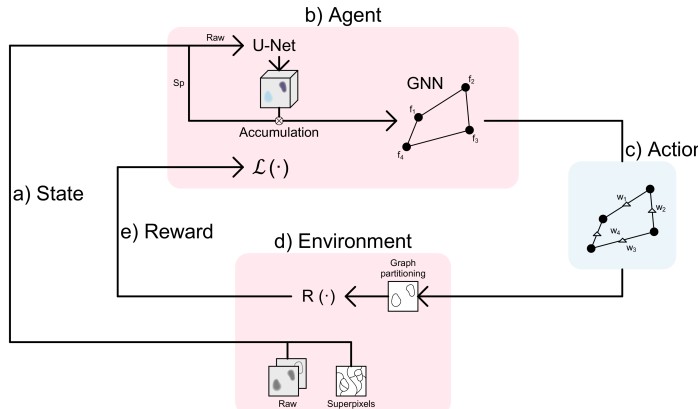

Figure 1: Interaction of the agent with the environment: (a) shows the state, which is composed of the image and superpixels; (b) depicts the agent, which consists of the actor and critic networks as well as the feature extractor that computes the node input features; (c) given the state, the agent performs the actions by predicting edge weights on the graph; (d) the environment, which includes the image, superpixels, graph and graph partitioning based on the weights predicted through agent actions ; (e) rewards are obtained by evaluating the segmentation arising from the graph partitioning, based on pre-defined and data dependent rules. The rewards are given back to the agent where they are used for training.

## 3 METHODS

The task of instance segmentation can be formalized as transforming an image $x$ into a labeling $y$ that maps each pixel to a label value. An instance corresponds to the maximal set of pixels with the same label value. Typically, the instance segmentation problem is solved via supervised learning, i.e. using a training set with groundtruth labels $\hat{y}$. Note that $y$ is invariant under the permutation of label values, which makes it difficult to formulate instance segmentation in a fully differentiable manner. Most approaches first predict a "soft" representation with a CNN, e.g. affinities (Gao et al. 2019; Lee et al. 2017; Wolf et al. 2020), boundaries (Beier et al. 2017; Funke et al. 2018) or embeddings (De Brabandere, Neven, and Van Gool 2017; Neven et al. 2019) and apply non-differentiable post-processing, such as agglomeration (Bailoni et al. 2019; Funke et al. 2018), density clustering (Comaniciu and Meer 2002; McInnes and Healy 2017) or partitioning (Andres et al. 2012), to obtain the instance segmentation. Alternatively, proposal-based methods predict a bounding-box per instance and then predict the instance mask for each bounding-box (He et al. 2017). Furthermore, the common evaluation metrics for instance segmentation (Meilă 2003; Rand 1971) are also not differentiable.

Our main motivation to explore RL for the instance segmentation task is to circumvent the restriction to differentiable losses and - regardless of the loss - to make the whole pipeline end-to-end even in presence of non-differentiable steps that transform pixelwise CNN predictions into instances.

We formulate the instance segmentation problem using a region adjacency graph $G = (V, E)$, where the nodes $V$ correspond to superpixels (clusters of pixels) and the edges $E$ connect nodes that belong to spatially adjacent superpixels. Given edge weights $W$, the instance segmentation is obtained by partitioning the graph, here using an approximate multicut solver (Kernighan and Lin 1970). Together, the image data, superpixels, graph and the graph partitioning make up the environment $\mathcal{E}$ of our RL setup. Based on the state $s$ of $\mathcal{E}$, the agent $\mathcal{A}$ predicts actions $a$. Here, the actions are interpreted as edge weights $W$ and used to partition the graph. The reward $r$ is then computed based on this partitioning. Our agent $\mathcal{A}$ is a stateless actor-critic (Haarnoja et al. 2018), represented by two graph neural networks (GNN) (Gilmer et al. 2017). The actor predicts the actions $a$ based on the graph and its node features $F$. The node(superpixel) features are computed by pooling together the corresponding pixel features based on the raw image data.

We compute the node features $F$ with a UNet (Ronneberger, Fischer, and Brox 2015) that takes the image as input and outputs a feature vector per pixel. These features are spatially averaged over the

superpixels to obtain $F$. The feature extractor UNet is part of the agent $\mathcal{A}$, thus training it end-to-end with the actor and critic networks (Fig. 1). In low data regimes it is also possible to use a pre-trained and fixed feature extractor or to combine the learned features with hand-crafted ones.

Crucially, the reinforcement setup enables us to use both a non-differentiable instance segmentation step and reward function, by encapsulation of the "pixels to instances" step in the environment and learning a policy based on the rewards with the stateless actor critic.

## 3.1 STATELESS REINFORCEMENT LEARNING SETUP

Unlike most RL settings (Sutton and Barto 2018), our approach does not require an explicitly time dependent state: the actions returned by the agent correspond to the real-valued edge weights in $[0, 1]$, which are used to compute the graph partitioning. Any state can be reached by a single step from the initial state and there exists no time dependency in the state transition. Unlike Jain et al. 2011, we predict all edge values at once which allows us to avoid the iterative strategy of Araslanov, Rothkopf, and Roth 2019 and deliver and evaluate a complete segmentation in every step. Hence, we implement a stateless actor critic formulation.

Stateless RL was introduced in Pfau and Vinyals 2016 to study the connection between generative adversarial networks and actor critics, our method is one of the first practical applications of this concept. Here, the agent consists of an actor, which predicts the actions $a$ and a critic, which predicts the action value $Q$ (expected future discounted reward) given the actions. The stateless approach simplifies the action value: it estimates the reward for a single step instead of the expected sum of discounted future rewards for many steps. We have explored a multi-step setup as well, but found that it yields inferior results for our application; details can be found in the App. A.8. Furthermore, we compute sub-graph rewards instead of relying on a single global reward in order to provide a more localized reward signal (see Section 3.2 for details).

The actor corresponds to a single GNN, which predicts the mean and variance of a Normal distribution for each edge. The actions $a$ are determined by sampling from this distribution and applying a sigmoid to the result to obtain continuous edge weights in the value range $[0, 1]$. The GNN takes the state $s = (G, F)$ as input arguments and its graph convolution for the $i^{th}$ node is defined as in Gilmer et al. 2017:

$$f_i = \gamma_\pi \left( f_i, \frac{1}{|N(i)|} \sum_{j \in N(i)} \phi_\pi \left( f_i, f_j \right) \right) \tag{1}$$

where $\gamma_\pi$ as well as $\phi_\pi$ are MLPs, $(\cdot, \cdot)$ is the concatenation of vectors and $N(i)$ is the set of neighbors of node $i$. The gradient of the loss for the actor is given by:

$$\nabla_\theta \mathcal{L}_{actor} = \nabla_\theta \frac{1}{|SG|} \sum_{sg \in G} \left[ \alpha \sum_{\hat{a} \in sg} log(\pi^\theta(\hat{a}|s)) - Q_{sg}(s, a) \right] \tag{2}$$

This loss gradient is derived following Haarnoja et al. 2018. We adapt it to the sub-graph reward structure by calculating the joint action probability of the policy $\pi^\theta$ over each sub-graph $sg$ in the set of all sub-graphs $SG$. Using this loss to optimize the policy parameters $\theta$ minimizes the Kullback-Leibler divergence between the Gibbs distribution of action values for each sub-graph $Q_{sg}(s, a)$ and the policy with respect to the parameters $\theta$ of the policy. $\alpha$ is a trainable temperature parameter which is optimized following the method introduced by Haarnoja et al. 2018.

The critic predicts the action value $Q_{sg}$ for each sub-graph $sg \in SG$. It consists of a GNN $Q_{sg}(s, a)$ that takes the state $s = (G, F)$ as well as the actions $a$ predicted by the actor as input and predicts a feature vector for each edge. The graph convolution from Equation 1 is slightly modified:

$$f_i = \gamma_Q \left( f_i, \frac{1}{|N(i)|} \sum_{j \in N(i)} \phi_Q \left( f_i, f_j, a_{(i,j)} \right) \right) \tag{3}$$

again $\gamma_Q$ and $\phi_Q$ are MLPs. Based on these edge features $Q_{sg}$ is predicted for each sub-graph via an MLP. Here, we use a set of subgraph sizes (typically, 6, 12, 32, 128) to generate a supervison signal

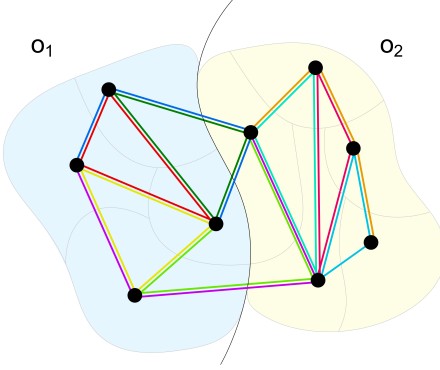

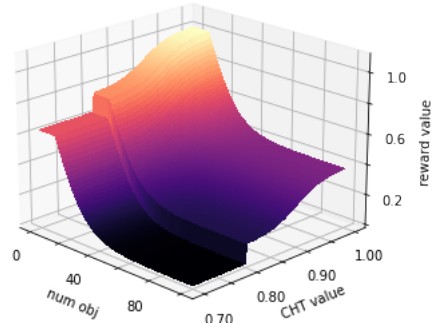

Figure 2: The graph is subdivided into sub-graphs, each sub-graph is highlighted by a different color. All sub-graphs have the same number of edges (here 3). Overall, we use a variety of sizes covering different notions of locality.

Figure 3: An example reward landscape Circle Hough Transform (CHT) rewards. High rewards are given if the overall number of predicted objects is not too high and if the respective object has a large CHT value.

for different neighborhood scales. A given MLP is only valid for a fixed graph size, so we employ a different MLP for each size. The loss for the critic is given by:

$$\mathcal{L}_{critic} = \frac{1}{|SG|} \sum_{sg \in G} \frac{1}{2} (Q_{sg}^{\delta}(s, a) - r)^2 \tag{4}$$

Minimizing this loss with respect to the action value function's parameters $\delta$ minimizes the difference between the expected reward and action values $Q_{sg}^{\delta}(s, a)$.

## 3.2 LOCALIZED SUB-GRAPH REWARDS

In most RL applications a global scalar reward is provided per state transition. In our application of graph-based instance segmentation, it is instead desirable to introduce several more localized rewards in order to learn from a reward for the specific action, rather than a global scalar. Here, reward decomposition is natural because we evaluate the segmentation quality per object and can use the object scores to provide a localized reward. In order to formalize this idea, we have designed our actor critic (Section 3.1) to learn from sub-graph rewards.

A good set of sub-graphs should fulfill the following requirements: each sub-graph should be connected so that the input to the MLP that computes the activation value for the sub-graphs is correlated. The size of the sub-graphs should be adjustable and all sub-graphs should be extracted with the exact same size to be valid inputs for the MLP. The union of all sub-graphs should cover the complete graph so that each edge contributes to at least one action value $Q_{sg}$. The sub-graphs should overlap to provide a smooth sum of action values. We have designed Alg. 1 to extract a set of sub-graphs according to these requirements. Fig. 2 shows an example sub-graph decomposition.

While some of the rewards used in our experiments can be directly defined for sub-graphs, most are instead defined per object (see App. A.2 for details on reward design). We use the following general procedure to map object-level rewards to sub-graphs: first assign to each superpixel the reward of its corresponding object. The reward per edge is determined by the maximum value of its two incident superpixels' rewards. The edge rewards are averaged to obtain the reward per sub-graph.

By taking the maximum we assign the higher score to edges whose incident superpixels belong to different objects, because they probably correspond to a correct split. Note that the uncertainty in the assignment of low rewards can lead to a noisy reward signal, but the averaging of the edge rewards over the sub-graphs and the overlaps between the sub-graphs smooth the rewards. We have also explored a different actor critic setup that can use object level rewards directly, with no sub-graph extraction and mapping. However, this approach yields inferior results, see App. A.3 for details.

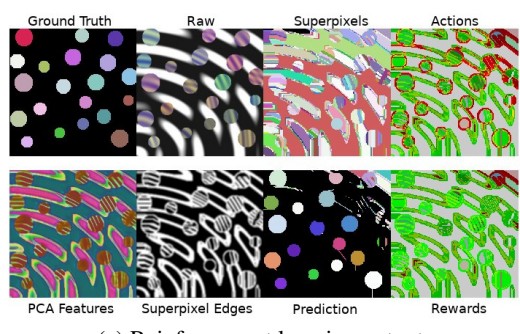 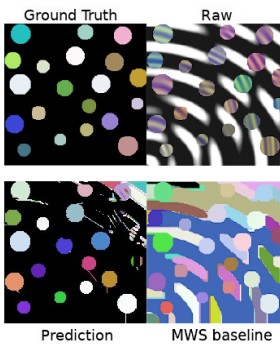

(a) Reinforcement learning output.  (b) Mutex watershed baseline.

Figure 4: Synthetic data. a) top left to right: groundtruth segmentation, raw data, superpixels and visualization of the actions (merge actions in green, split actions in red). Bottom left to right: pre-trained pixel embeddings, superpixel edges, segmentation result and visualization of the rewards (light green for high rewards, dark red for low rewards. b) comparison of segmentation from our method and the mutex watershed.

## 4 EXPERIMENTS

We evaluate our approach on three instance segmentation problems: one synthetic and two real. For a proof-of-principle, we construct a synthetic dataset with circular shapes on structured background, showing how our framework can exploit simple geometric priors. Next, we apply the method to a popular microscopy benchmark dataset for nucleus segmentation (Caicedo et al. 2019). Finally, we consider a challenging biological segmentation problem with boundary-labeled cells. Here, we evaluate both learning restricted to prior rules and mixed supervision combining rule-based and direct rewards computed from groundtruth annotations. The problem setup, network architectures and hyperparameters are reported in detail in App. A.9.

### 4.1 SYNTHETIC DATA: CIRCLES ON STRUCTURED GROUND

We create synthetic images of circles on a structured background and segment this data using only simple geometric rules. Superpixels were generated with the mutex watershed (Wolf et al. 2020) applied to the gaussian gradient of the image. Here, we demonstrate that the actor critic can be trained without any direct object-level supervision and apply a simplified setup with a fixed pixel feature extractor, pre-trained through self-supervision (see App. A.1).

The object-level reward is based on the Circle Hough Transform (CHT) (Hassanein et al. 2015). It is combined with an estimate for the total number of objects in the image as an additional global reward. The global reward gives useful gradients during early training stages: when too few potential objects are found in the prediction, a low reward can be given to the tentative background object. If too many potential objects are found, a low reward can be given to all the foreground objects with a low CHT value. The surface created by the per-object and global reward is shown in Fig. 3. The exact reward computation can be found in App. A.11.

Fig. 4 shows the output of all algorithm components on a sample image. We also computed results with the mutex watershed (Wolf et al. 2020), a typical algorithm for boundary based instance segmentation in microscopy. Texture within objects and structured background are inherently difficult for region-growing algorithms, but our approach can exploit higher-level reasoning along with low-level information and achieve a good segmentation.

### 4.2 REAL DATA: NUCLEUS SEGMENTATION

Nuclei are a very frequent target of instance segmentation in microscopy, which is also reflected in the large amount of publicly available annotated data. The availability of training data sparked the development of popular pre-trained solutions, such as a generalist UNet (Falk et al. 2019) or StarDist

(Schmidt et al. 2018) and CellPose (Stringer et al. 2021) which both have an (implicit) shape prior. Also, due to ubiquity of nuclei in microscopy, detailed prior knowledge exists on their shape and their appearance under different stainings. The experiments in this section aim to answer the following questions: i) given fully annotated groundtruth images for training, is there an advantage in using our RL formulation with object-level rewards compared to commonly used fully supervised baselines? ii) given superpixels that can be combined into the correct solution, but no other direct supervision, can our approach learn to combine the superpixels correctly only from high-level rules? iii) what happens if superpixels are suboptimal? For data, we turn to the dataset of Caicedo et al. 2019 and select images that contain nuclei of medium size (175 for training and 22 for test).

Features are learned end-to-end. In the unsupervised setting, we compute the reward by combining several object descriptors: eccentricity, extent, minor diameter, perimeter, solidity as well as mean, maximum and minimum intensity per object. The object reward is then given by the normalized sum of square distances of these quantities and their expected value. Objects larger than 15,000 pixels are considered to belong to the background and are not assigned a reward. Since the superpixels serve as fixed input into our model that do not get modified, the accuracy of our segmentations is bound by their accuracy. To investigate their influence, we evaluate our approach with three different sets of superpixels: "GT", where we intersect the superpixels with the groundtruth object masks to ensure that a correct segmentation can be recovered, "UNET", where we compute the superpixels using predictions of a pre-trained U-Net as an edge detector and "RAW", where we only take into account the raw image data. See A.12 for more details on superpixels and object descriptors.

| Method | Superpixel | mAP | IoU50 | IoU75 |
|---|---|---|---|---|
| UNet | - | 0.710 | 0.900 | 0.756 |
| StarDist | - | 0.645 | 0.938 | 0.736 |
| Cellpose | - | 0.666 | 0.931 | 0.776 |
| UNet + MC | GT | 0.674 | 0.806 | 0.702 |
| ours (sup.) | GT | **0.766** | 0.907 | 0.799 |
| Otsu | - | 0.554 | 0.763 | 0.579 |
| ours (unsup.) | GT | **0.743** | 0.916 | 0.787 |
| ours (unsup.) | UNET | 0.671 | 0.872 | 0.704 |
| ours (unsup.) | RAW | 0.453 | 0.785 | 0.439 |
| sp gt | UNET | 0.793 | 0.98 | 0.852 |
| sp gt | RAW | 0.554 | 0.969 | 0.505 |

Table 1: Nuclei segmentation: for mAP and IoU higher values are better. Methods above the first middle line were trained fully supervised. Methods below the first middle line were trained without groundtruth, the results below the second middle line indicate the quality of the superpixels projected to the groundtruth (best possible result that can be achieved with the given superpixels).

Tab. 1 summarizes the results, with a comparison to popular generalist pre-trained nuclear segmentation methods: StarDist (Schmidt et al. 2018), Cellpose (Stringer et al. 2021) and UNet (Falk et al. 2019). For StarDist and Cellpose, we use the pre-trained models provided with the papers. The UNet is trained on the same images as StarDist, the instance segmentation is recovered either by applying connected components to the boundary-subtracted foreground prediction ("UNet") or, to obtain a comparison conditioned on a particular set of superpixels, by using the UNet boundary predictions and superpixels described above as input to Multicut graph-based agglomeration ("UNet + MC"). Otsu threshold serves as a simple unsupervised baseline (Otsu 1979), where binarizing the image is followed by connected components to obtain the instance segmentation.

For the first question, we train our pipeline fully supervised ("ours (sup.)") as described in App. A.10: we use pixelwise groundtruth, but can also exploit our RL formulation where the loss is assigned to individual objects through the non-differentiable graph agglomeration step. Here, our method performs better than all baselines without RL, so there is clearly an advantage to using object-level supervision (as also demonstrated recently for non-RL setups, e.g. by Wolny, Yu, et al. 2021).

For the other two questions, we train our method using only rule-based rewards ("ours (unsup.)"). Given superpixels from which the groundtruth image can be recovered ("GT"), we then achieve better segmentation quality than the fully supervised baselines and the gap in performance between our unsupervised and supervised approach is smaller than the gap to the runner-up baseline. Of note, our unsupervised model also outperforms the "UNet + MC" baseline using the same "GT" superpixels, so its performance cannot be explained just by the use of groundtruth in superpixel generation. Example results and failure cases are shown in App. A.12.

In the third experiment, we use a pretrained UNet as an edge detector to create superpixels of "medium" quality and again obtain strong results, outperforming StarDist, CellPose and UNet+MC with "GT" superpixels. Finally, with our worst set of superpixels obtained directly from the raw data, the method can learn to exploit the rules, but is clearly hindered by the superpixel quality.

## 4.3 REAL DATA: CELL SEGMENTATION

Biomedical applications often require segmentation of objects of known morphology arranged in a regular pattern (Thompson 1992). Such data presents the best use case for our algorithm as the reward function can leverage such priors. We address a cell segmentation problem from developmental biology, where cells often follow stereotypical shapes and arrangement: 317 drosophila embryo images from Bhide et al. 2020, including 10 with expert annotations used as test data. Note that several pre-trained networks are available for cell segmentation in light microscopy (Chamier et al. 2021; Stringer et al. 2021; Wolny, Cerrone, et al. 2020); however, they produce sub-par results on this data due to experimental differences.

The rewards express that the cells are arranged in a radial pattern, with the average size known from other experiments (see Fig. 5). We set a high reward for merging superpixels that certainly belong to the background (close to the image boundary or center). For background edges near the foreground area, we modulate the reward by the circularity of the overall foreground contour. For the likely foreground edges, we compute object-level rewards by fitting a rotated bounding box to each object and comparing its radii and orientation to template values. We use a weight profile based on the known embryo width to combine object and background rewards (App. A.6).

| Method | VI | VI merge | VI split |
|---|---|---|---|
| sp gt | 1.266 | 0.672 | 0.594 |
| ours | 2.213 | 0.839 | 1.374 |
| ours (semisup.) | **1.634** | 0.733 | 0.901 |
| ours (handcrafted) | 2.523 | 0.987 | 1.536 |
| UNet + MC | 3.361 | 3.019 | 0.342 |
| contrastive | 4.440 | 1.155 | 3.28 |

Table 2: Cell segmentation results, measured by variation of information (VI) (Meilă 2003). This entropy-based metric is commonly used to evaluate crowded segmentations in microscopy. We also report its merge and split component that measure the over-/under-segmentation error respectively. Lower values are better.

More formally, the rewards are calculated as follows: for each edge, we define the position $h$ as the average of the centers of the two incident superpixels. Given the image center $c$, the radius of a circle that approximately covers the foreground $j$ and the (maximal) image border position $m$, we use a gaussian kernel $\mathcal{K}(\cdot)$ for weighting and define edge reward $r_{edge}$:

$$r_{bg} = \begin{cases} \mathcal{K}\left(\frac{||h-c||}{\gamma}\right)(1-a), & \text{if } h \leq j \\ \mathcal{K}\left(\frac{||m-h||}{\eta}\right)(1-a), & \text{otw} \end{cases} \quad (5)$$

$$r_{fg} = \mathcal{K}\left(\frac{||h-j||}{\delta}\right)\max(r_{o1}, r_{o2}) \quad (6)$$

$$r_{edge} = r_{fg} + r_{bg} \quad (7)$$

Here $\gamma$, $\eta$ and $\delta$ are normalization constants. The kernel function in Eq. 5 determines the background probability of an edge; $1 - a$ constitutes a reward that favors merges. It is scaled by the background probability. The object rewards $r_o$ are found by fitting a rotated bounding box to the object and then comparing orientation and extent to expected values known from previous experiments. They are mapped to edge rewards $r_{o1}, r_{o2}$ using the maximum value of the two incident objects.

We pre-compute superpixels by using boundary predictions as input to a watershed seeded from local maxima. We use the UNet from Wolny, Cerrone, et al. 2020, which was trained on roughly similar images. As it was trained on plant cells in different microscope modality, its prediction is far from perfect, especially around the inner circle, see Fig. 5"Edge prediction". We combine the learned node features with hand-crafted features: the normalized polar coordinate of the superpixel center and the normalized superpixel size. Fig. 5 shows visualisations of the learned and hand-crafted features. Interestingly, the learned features converge to a representation that resembles a semantic segmentation of boundaries.

Tab. 2 shows the results: "ours" is the method described above; for "ours (semisup.)" we train a model that additionally receives direct supervision from groundtruth for a single patch using the reward from App. A.10 and for "ours (handcrafted)" we only use the hand-crafted features and not the learned features. We include the UNet from Wolny, Cerrone, et al. 2020 with Multicut for instance segmentation ("UNet + MC") as well as the method of De Brabandere, Neven, and Van Gool 2017 trained on the same data as Wolny, Cerrone, et al. 2020 ("contrastive") as baselines. Since only 10 images of the dataset are annotated, we cannot efficiently finetune any of the popular cell segmentation networks on this dataset. We also project the superpixels to their respective groundtruth

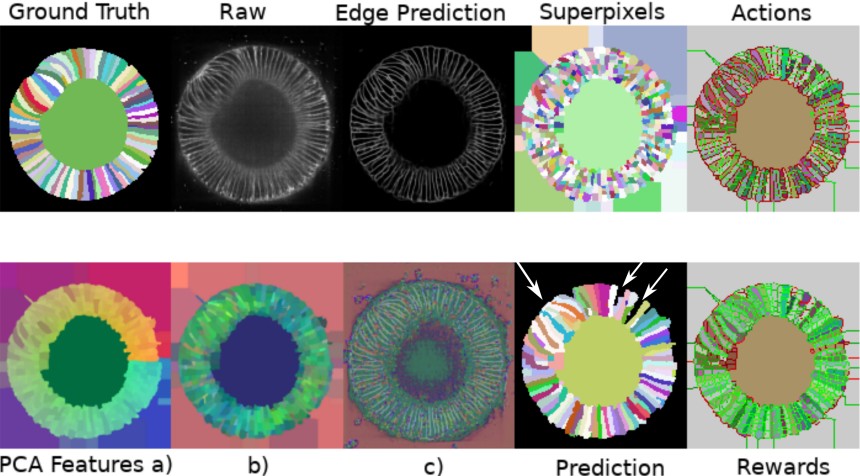

Figure 5: Cell segmentation experiment. Top left to right: groundtruth segmentation; raw data; boundary predictions; superpixel over-segmentation; visualization for the actions on every edge (green = merge action, red = split action). Bottom left to right: a) handcrafted features per superpixel; b) learned features averaged over superpixels; c) learned features per pixel; Multicut segmentations; visualization of the rewards (light green = high reward, dark red = low reward). For all features, we use the first 3 PCA components for visualisation. White arrows point to remaining errors.

cluster ("sp gt") to indicate the best possible solution that can be achieved with the given superpixels. Our approach clearly outperforms the baseline methods trained on the data from Wolny, Cerrone, et al. 2020. While predictions are not perfect (white arrows in Fig. 5, prior rules turn out to be sufficient to assemble most cells correctly. The remaining errors are caused by objects not fully conforming to the priors ("bent" rather than straight oval cells) or by a very weak boundary prediction. Furthermore, we see that the learned features significantly improve results and that the semi-supervised approach provides a large boost, even with a single patch used for direct supervision. We only report results for the best model as measured by the reward on a validation set across several training runs. App. Fig. 8 shows validation reward curves consistently improve during training for all random seeds.

## 5 DISCUSSION AND OUTLOOK

We introduced an end-to-end instance segmentation algorithm that can exploit non-differentiable loss functions and high-level prior information. Our novel RL approach is based on the stateless actor-critic and predicts the full segmentation at every step, allowing us to assign rewards to all objects and reach stable convergence. The segmentation problem is formulated as graph partitioning; we design a reward decomposition algorithm which maps object- and image-level rewards to sub-graphs for localized supervision. Our experiments demonstrate good segmentation quality on synthetic and real data using *only* rule-based supervision without any object- or pixel-level labels, such as centers, boxes or masks. Furthermore, in case of full supervision, our method enables end-to-end instance segmentation with direct object-level reasoning, which will allow for post-processing-aware training of segmentation CNNs. In the future, we plan to explore other tasks and reward functions and will further study the semi-supervised setup that showed very promising initial results.

**Limitations** Our method relies on superpixels which are fixed and not optimized jointly, so the upper bound on the performance is defined by superpixel accuracy. We believe an extension to pixels is possible and envision working on this in the future, but the current setup will not scale to the pixel level directly. Also, our method is limited to problems where consistent prior rules can be formulated for all instances. While this is the case for many applications in life science and medical imaging, not all object classes in the natural world can be described in this way. Here, our method could contribute by complementing training annotations with rules, reducing the overall labelling load in a semi-supervised setting. Finally, our approach requires non-trivial reward engineering as a trade-off for not performing any annotations.

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
