# OpenReview forum: "Reinforcement learning for instance segmentation with high-level priors"
_ICLR.cc/2023/Conference — Submitted to ICLR 2023_

### Official Review · Reviewer_KFcT · 2022-10-24

**Confidence:** 4
**Correctness:** 3
**Technical Novelty And Significance:** 3
**Empirical Novelty And Significance:** 3
**Recommendation:** 5

**Clarity, Quality, Novelty And Reproducibility:**

The paper is generally clearly written. However, the clarity of the methods could be improved. See the weakness above.
There lack extensive experiments on public instance segmentation benchmark datasets,
The novelty is limited due to lacking the comparisons on more complicated medical dataset.
The experiments can hardly reproduced because the lacking of detials on superpixel graphs, which is the determine factor to the algorithm.


**Strength And Weaknesses:**

Strength:

+Training model with non-differentiable factors. When dealing with limited samples, the reinforcement learning plays a key role and the author take the advantage of it.

+The experiments is interesing and complete. They are conducted on both synthic and real images.

Weakness:

-The reason for utilizing the superpixel as the prior is not well explained.
-The estimation to computatial cost and inference speed (e.g., FLOPs, FPS) are not presented in the experiments.

-Comparisons to standard instance segmentation are missing. Although the algorithm has the advantage in limited samples, it doesn't provide enough avidence to demonstrate the compatitive scores when the training samples are adquated

-Experiments on real dataset is too simple to indicates the potential ability. Authors should provide the results on more complicated medical images.

**Summary Of The Paper:**

This paper explores the instance segmentation via actor-critic reinforcement learning. Instead of large amount of object-level supervision, the author formulate the instance segmentation as graph partitioning and predict the edge weights driven by the reward from object shape, position and size. Experiments on toy and real data demonstrate the effectiveness of the methods with a good set of priors.
Overall the proposed algorithm provides a new view regarding prior knowledge embedding, which enable well trained model with less annotated labels.

**Summary Of The Review:**

This paper address the limited labels for instance segmentation thorough actor-critic reinforcement learning. It follows the standard RL framework and take the object shape, position and size as the prior. By assigning the reward for each actor, the optimal instance score is obtained.
Overall the paper is well origanized but the experiments are not adquate to demonstrate the effectiveness of the algorithm. The reviewer suggest the author conduct more experiments on mote compilicated benchmark for instance segmentation.

---

### Official Review · Reviewer_qRMW · 2022-10-25

**Confidence:** 2
**Correctness:** 3
**Technical Novelty And Significance:** 3
**Empirical Novelty And Significance:** 2
**Recommendation:** 5

**Clarity, Quality, Novelty And Reproducibility:**

Can you provide an evaluation of the quality, clarity, and originality of the work?
Clarity: the paper is well-presented and the experiments are clear.
Quality: Please refer to the strengths.
Originality: the paper brings prior knowledge into end-to-end learning via using RL
Reproducibility: we’ve known that RL is hard to train, moreover, with GNN in the MDP, in my opinion, it seems hard to make the model converge. I recommend the authors provide more implementation details.


**Strength And Weaknesses:**

Strength:
The paper is well-motivated and well-written.
The overall RL environment looks sound and reasonable.
The experiments show the feasibility of the method.

Weakness:
The first paragraphs of the section 3 method, which, in my opinion, tries to describe the RL formulation of the problem, is hard to grasp overall, especially on how the image is segmented by graph partition.
Although the experiments show the feasibility of the approach, the method is quite limited to segmentation problems with strong priors. Therefore, the potential of expanding the idea to more real-world datasets is quite unclear, thus, it makes the contribution limited.


**Summary Of The Paper:**

- The paper formulates instance segmentation as a RL problem based on a stateless actor-critic setup, achieving end-to-end learning while exploiting prior knowledge on instance appearance instead of using ground truth instance supervision.
- The paper introduces a strategy for spatial decomposition of rewards based on fixed-sized subgraphs to enable localized supervision from combinations of object- and image-level rules.
- The paper demonstrates the feasibility of the approach on synthetic and real images and delivers excellent segmentations without ground truth supervision.


**Summary Of The Review:**

The paper is quite novel but the contribution is fairly limited.

---

### Official Review · Reviewer_qqMa · 2022-10-25

**Confidence:** 4
**Clarity, Quality, Novelty And Reproducibility:** I do not think it is necessary to add…
**Correctness:** 2
**Technical Novelty And Significance:** 3
**Empirical Novelty And Significance:** 2
**Recommendation:** 5

**Strength And Weaknesses:**

Quality/Clarity: the paper is well written and the techniques presented are easy to follow. Its motivation to use reinforcement learning for instance segmentation is clear, such as the non-differentiable step of instance extraction and lack of annotated training data.

Originality/significance: the idea is interesting, and there are multiple strengths:  (1) superpixels and graph neural network feature exaction (2) sub-graph reward mapping. However, it is a little weird to add reinforcement learning here. Additionally, the authors claim that there is no state transition. Without the buzzword "reinforcement learning", I think we can achieve similar results using the designs (GNN+reward mapping) in this paper.

**Summary Of The Paper:**

This paper formulates the instance segmentation problem as a graph partitioning problem, and uses reinforcement learning to predict the edge weights, which are the inputs to multicut solver for object segmentation. The authors introduce a strategy for spatial decomposition of rewards based on fixed-sized subgraphs to generate the object-to-sub-graph reward mapping to guide the action prediction. The experiments on toy and real data demonstrate that a good set of priors is sufficient to reach excellent performance without any direct object-level supervision.

**Summary Of The Review:**

The paper use actor critic to predict the edge weights and then use graph partition for instance segmentation. Overall it is a good paper, but not ready for publication.

---

> ### Author Response · Authors · 2022-11-10
> **Discussion on why we need reinforcement learning**
>
> We thank the reviewer for the suggestion to consider a simplified set-up without RL. We believe this could indeed work if we were addressing a fully supervised problem, like we do to create the baseline “ours (sup.)” in Table 1. In this case, we’d have direct groundtruth for the edges which we could map on the GNN to drive the training. However, our real application is to not use any direct supervision, but do prior-based rewards which can be derived from object-level or image-level prior knowledge. These rewards get mapped to subgraphs in order to provide a localized reward signal and learn a segmentation policy using an actor-critic setup. Our actor predicts edge weights (in the range from -infinity to infinity) that are transformed into a segmentation using the Multicut graph partitioning. We do not know how to provide a training signal to this network other than using an indirect RL based approach, since we cannot obtain target edge values from our reward mapping scheme that could be directly optimized. While we do obtain per edge reward values in our reward mapping approach, these cannot be directly used to regress or otherwise directly optimize the edge weights predicted by the actor. We therefore use the critic to learn an approximation of the loss.
>
> “Additionally, the authors claim that there is no state transition.” Indeed, we use a stateless RL formulation. We found that this setup matches our problems better than a state-based approach, see Appendix A.8 for discussions about alternative state-based design solutions we have pursued. We do not see the use of a stateless setup as a drawback of our method, and rather think that it provides one of the first real-world applications of the concept of a stateless actor-critic setup that was introduced for theoretical analysis by Pfau&Vinyals in https://arxiv.org/abs/1610.01945. Remarkably, they discuss a connection of actor-critic to GANs, which can be viewed as a different, much more wide-spread, approach to enable training of a network without direct application of a closed-form loss.

---

### Decision · Program_Chairs · 2023-01-20

**Decision:**

Reject

**Justification For Why Not Higher Score:**

The limited scope of the contribution and the corresponding evaluation is the primary reason that acceptance is not recommended.

**Justification For Why Not Lower Score:**

N/A

**Metareview: Summary, Strengths And Weaknesses:**

All reviewers for this paper concurred that this paper presents an interesting method. However, reviewers expressed concerns about the relatively limited scope of the contribution and the corresponding evaluation. There was also some question of whether the reinforcement learning-based approach makes the most sense and provides sufficient value given the additional complexity added. Overall, all reviewers concurred that the paper in the current form, while interesting, is below the acceptance threshold for ICLR. After reviewing the paper and author and reviewer comments, I agree with the shared assessment of the reviewers and do not recommend acceptance of this paper at this time.